# Cytokinin regulates vegetative phase change in *Arabidopsis thaliana* through the miR172/TOE1-TOE2 module

Sören Werner [1], Isabel Bartrina [1,2] & Thomas Schmülling [1✉]

During vegetative growth plants pass from a juvenile to an adult phase causing changes in shoot morphology. This vegetative phase change is primarily regulated by the opposite actions of two microRNAs, the inhibitory miR156 and the promoting miR172 as well as their respective target genes, constituting the age pathway. Here we show that the phytohormone cytokinin promotes the juvenile-to-adult phase transition through regulating components of the age pathway. Reduction of cytokinin signalling substantially delayed the transition to the adult stage. *t*Z-type cytokinin was particularly important as compared to iP- and the inactive *c*Z-type cytokinin, and root-derived *t*Z influenced the phase transition significantly. Genetic and transcriptional analyses indicated the requirement of SQUAMOSA PROMOTER BINDING PROTEIN-LIKE (SPL) transcription factors and miR172 for cytokinin activity. Two miR172 targets, *TARGET OF EAT1* (*TOE1*) and *TOE2* encoding transcriptional repressors were necessary and sufficient to mediate the influence of cytokinin on vegetative phase change. This cytokinin pathway regulating plant aging adds to the complexity of the regulatory network controlling the juvenile-to-adult phase transition and links cytokinin to miRNA action.

[1] Institute of Biology/Applied Genetics, Dahlem Centre of Plant Sciences (DCPS), Freie Universität Berlin, Albrecht-Thaer-Weg 6, 14195 Berlin, Germany. [2] Present address: Institute of Biology, University of Graz, Schubertstrasse 51, 8010 Graz, Austria. ✉email: t.schmuelling@fu-berlin.de

Flowering plants progress through juvenile and adult phases of vegetative development before undergoing a transition to reproductive growth. The proper timing of these phases strongly influences plant fitness and reproductive success. Phase transitions are controlled by exogenous factors such as light (period and intensity) and temperature, as well as endogenous cues such as carbohydrate assimilates (mainly sucrose), hormones (e.g., gibberellin (GA)) and plant age feeding into intrinsic genetic programs[1–3].

The transition from the juvenile to the adult vegetative phase is characterized by changes of shoot morphology as well as an increase in reproductive potential. In *Arabidopsis thaliana*, vegetative phase change is accompanied by an increase in size and length/width ratio of the leaf blade, an increase in the degree of serration of the leaf margins and a decrease in cell size[4,5]. In addition to these gradual changes, the appearance of trichomes on the abaxial side of leaves has been commonly used to define the transition from the juvenile to the adult phase[6].

Two evolutionary highly conserved miRNAs, miR156 and miR172, and their respective target genes are at the center of the genetic mechanisms regulating the juvenile-to-adult phase transition[7,8]. The precursors of these miRNAs are encoded by ten (*MIR156A-J*) and five genes (*MIR172A-E*), respectively[9,10]. As the plant ages, the level of miR156 and the partly redundant miR157 decreases[11–13]. They target ten out of 16 *SQUAMOSA PRO-MOTER BINDING PROTEIN-LIKE* (*SPL*) genes, a group of transcription factor genes controlling several aspects of *Arabidopsis* shoot development, such as the timing of juvenile-to-adult and vegetative-to-reproductive phase transitions, leaf initiation rate, and floral organ development[14–16]. In contrast to miR156, miR157 has only a minor effect on *SPL* expression and shoot morphology[13]. MiR156/miR157 decline with progressing plant development results in an increment of *SPL* mRNA abundance and translation[14,17]. Since at least five out of the ten miR156/miR157-targeted *SPL* genes are directly involved in promoting the transcription of *MIR172* genes[14], the abundance of miR172 increases with plant age[18]. The targets of miR172 include the floral organ identity gene *APETALA2* (*AP2*) and the AP2-like genes *SCHLAFMÜTZE* (*SMZ*), *SCHNARCHZAPFEN* (*SNZ*), *TARGET OF EAT1* (*TOE1*), *TOE2*, and *TOE3*, which encode transcriptional regulators known to act as repressors of vegetative phase change and transition to flowering[18–22]. Corresponding to their opposite expression patterns, miR156 and *AP2-like* genes promote the juvenile phase, whereas miR172 and *SPL* genes promote the transition to the adult and the reproductive phase, as well as the respective accompanying heteroblastic features[13,23].

In transgenic plants overexpressing miR172 abaxial trichomes are produced earlier, whereas abaxial trichome formation is delayed in miR172 knockout mutants with *MIR172A* and *MIR172B* playing dominant roles in the regulation of epidermal identity[11,24]. Leaf shape however is morphologically inconspicuous in plants with altered miR172 levels, indicating that in contrast to miR156/SPL, the miR172/AP2-like module affects only a subset of the leaf traits changing in the course of the transition[11]. Among the miR172 target genes, *TOE1* and *TOE2* have a very strong impact on juvenile leaf number: knocking out one of them already decreases the number of leaves without abaxial trichomes significantly, and simultaneous loss of both genes reduces the number of juvenile leaves even to less than half compared to the wild type[11]. Not much is known about the degree of individual influence of the other four target genes (*AP2*, *SMZ*, *SNZ*, and *TOE3*), but all six target genes of miR172 have to be knocked out in order to phenocopy a miR172 overexpressor[25].

The phytohormone cytokinin (CK) plays a role in multiple processes that influence the growth and development of root and shoot organs. Amongst others, CK regulates cell proliferation and differentiation, the size and activity of apical meristems, photomorphogenesis, apical dominance, phyllotaxis, flowering time, and leaf senescence[26]. CK regulates also numerous processes related to nutritional cues and biotic and abiotic stress responses[27].

CKs are $N^6$-substituted adenine derivatives whose metabolism and signal perception and transduction are largely known[28]. Isopentenyltransferases (IPTs) catalyze the formation of iso-pentenyladenine (iP) and *cis*-zeatin (*cZ*) ribotides, the former of which can be hydroxylated by the cytochrome P450 mono-oxygenases CYP735A1 and CYP735A2 to form *trans*-zeatin (*tZ*) ribotides[29–32]. These precursors are converted into the corresponding bioactive free bases by the LONELY GUY (LOG) phosphoribohydrolases[33,34]. Reduction of the levels of active CKs is achieved through irreversible degradation by CK OXIDASE/DEHYDROGENASES (CKXs) or conjugation to sugar moieties, most commonly glucose, rendering them inactive[35,36]. The CK signal transduction pathway is a multi-step His-Asp phosphorelay by a two-component signaling system[28]. In *Arabidopsis*, three membrane-bound histidine kinases serve as CK receptors: ARABIDOPSIS HISTIDINE KINASE2 (AHK2), AHK3, and CYTOKININ RESPONSE 1 (CRE1), also known as AHK4. CK binding to the receptor proteins results in an autophosphorylation of their kinase domains and the intramolecular transfer of the phosphoryl group to the receiver domain[37–40]. Histidine phosphotransfer proteins (AHPs) shuttle the phosphoryl group from the cytosol to the nucleus, where they activate type-B response regulators (ARRs). Acting as transcription factors, the type-B ARRs mediate the transcriptional output of the CK response. Among the primary response genes are type-A *ARR* genes encoding feedback inhibitors of the CK signaling pathway[41,42].

Little is known about the influence of phytohormones on the juvenile-to-adult phase transition except for GA. GA was shown to promote the expression of some *SPL* genes and *MIR172B* and positively influence vegetative phase change[6,43–47]. Here, we describe a role for CK in the age-dependent regulation of vegetative phase change. We identified the most important components of the CK signaling pathway involved, as well as the convergence point with the age pathway. CK promotes the expression of miR172 genes, most probably involving SPL proteins. The miR172 targets TOE1 and TOE2 are necessary and sufficient to mediate the effect of CK on the juvenile-to-adult phase transition. Overall, this work describes the molecular basis for the influence of CK on vegetative phase change regulated by the age pathway.

## Results

**Cytokinin is a positive regulator of vegetative phase change**. In *Arabidopsis*, the first rosette leaves are small, unserrated and round-shaped. With advancing age, leaf blade size and serration increase with every newly formed leaf[23]. Comparing the leaf morphology of mutants with a lower CK status like the CK receptor mutant *ahk2 ahk3* and plants overexpressing the CK-degrading enzyme CKX1 (CKX1ox) with wild type, we observed a larger number of leaves with juvenile features (Fig. 1). In order to investigate the possible influence of CK on vegetative phase change, we examined the appearance of abaxial trichomes in CKX1ox and *ahk2 ahk3*, as well as in *ckx3,4,5,6* and the *AHK2* gain-of-function mutant *rock2*[48] as plants with an increased CK content or signaling, respectively. In addition, we investigated a possible dependence of the phenotype on photoperiod since a long photoperiod stimulates abaxial trichome formation[46]. Plants were grown under both long-day (LD; 16 h of light, 8 h of darkness) and short-day conditions (SD; 8 h of light, 16 h of

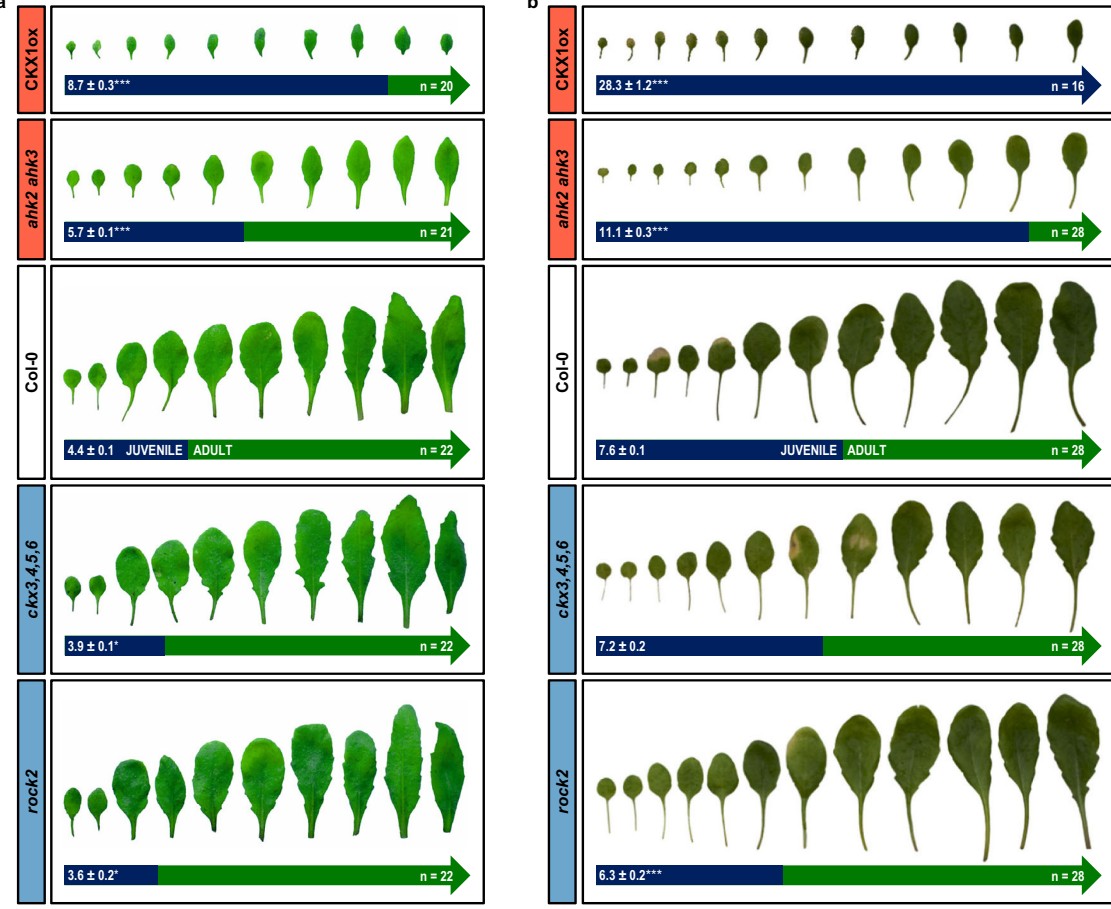

**Fig. 1 Cytokinin promotes the juvenile-to-adult phase transition in a photoperiod-independent manner in *Arabidopsis*. a** Morphology of the first ten leaves of *Arabidopsis* lines with an altered CK status grown in LD. **b** Morphology of the first 12 leaves of *Arabidopsis* lines with an altered CK status grown in SD. The color of the arrow indicates juvenile leaves (blue) and adult leaves (green). Values indicate the mean number of leaves without abaxial trichomes ± SEM. Asterisks indicate significant differences compared to the wild type of the respective experiment, as calculated by Kruskal–Wallis test (*$q < 0.05$; **$q < 0.01$; ***$q < 0.001$). Exact $q$-values are included in the Source Data files.

darkness). We observed the same positive correlation of the altered CK status and the onset of the adult phase under both photoperiods, with more pronounced phenotypic differences under SD. Genotypes with a higher CK status displayed an earlier transition to the adult phase, whereas a reduced CK status prolonged the juvenile phase (Fig. 1).

Among the tested genotypes, the earliest transition to the adult phase was observed in the *rock2* mutant: under SD conditions *rock2* plants produced an average of 6.3 ± 0.2 juvenile leaves compared to 7.6 ± 0.1 juvenile leaves in wild type. Under LD conditions these were 3.6 ± 0.2 and 4.4 ± 0.1 juvenile leaves for these two genotypes. In contrast, *ahk2 ahk3* plants formed 11.1 ± 0.3 and CKX1ox plants even 28.3 ± 1.2 leaves without abaxial trichomes under SD conditions. Ahk2 ahk3 and CKX1ox plants grown under LD conditions generated 5.7 ± 0.1 and 8.7 ± 0.3 juvenile leaves, respectively (Fig. 1).

Taken together, the results showed a positive influence of CK on the transition from the juvenile to the adult vegetative phase, which was photoperiod-independent.

**CK genes involved in the regulation of vegetative phase change**. In order to determine which CK genes of *Arabidopsis* might be functionally relevant, a series of mutants with altered CK metabolism or signaling causing a lower or higher CK status was analyzed for the appearance of abaxial trichomes under SD conditions. Among the mutations affecting CK biosynthesis and

transport, a delay of the vegetative phase transition was only observed when the biosynthesis or the transport of *t*Z-type CK was affected (*ipt3,5,7*; *cypDM*; *abcg14*) whereas plants lacking *c*Z-type CK (*ipt2,9*) behaved similar to wild type (Table 1). Mutations in the biosynthesis genes *LOG4* and *LOG7*, which are expressed in the shoot apical meristem (SAM) and most likely provide the active CK for normal shoot meristem regulation[49,50], did not alter juvenile leaf number (Table 1). Interestingly, plants lacking three *LOG* genes (*log3,4,7*) showed a tendency to a shorter juvenile phase (Table 1), contrasting the other results. Loss of four *CKX* genes, including the SAM-expressed *CKX3* and *CKX5*[51], resulted in a weak, but reproducible reduction in juvenile leaf number (Fig. 1 and Table 1), whereas the *ckx3 ckx5* double mutant did not show any changes compared to wild-type plants (Table 1).

Among the three receptor double mutants only *ahk2 ahk3* displayed a prolonged juvenile phase compared to the wild-type control, whereas *ahk2 cre1* and *ahk3 cre1* behaved like wild type (Fig. 2a). This suggests that AHK2 and AHK3 together control the CK-dependent juvenile-to-adult phase transition. Interestingly, among the mutants harboring a constitutively active cytokinin receptor, *rock2* but not *rock3* produced a reduced number of juvenile leaves (Fig. 2a), indicating that AHK2 might be more relevant than AHK3 to mediate the vegetative phase transition.

There are 11 type-B *ARR* genes in *Arabidopsis* specifying the transcriptional CK response downstream of the receptors. To test

**Table 1 Number of juvenile leaves of wild-type and cytokinin mutant plants grown in short days.**

| | No. of leaves w/o abaxial trichomes | n |
|---|---|---|
| *Experiment 1* | | |
| Col-0 | 11.2 ± 0.3 | 24 |
| *log4* | 11.4 ± 0.4 | 25 |
| *log4 log7* | 10.8 ± 0.4 | 25 |
| *ipt2 ipt9* | 11.8 ± 0.4 | 25 |
| *ipt3,5,7* | 12.9 ± 0.3** | 24 |
| *abcg14* | 13.3 ± 0.3*** | 25 |
| *cypDM* | 15.5 ± 0.3*** | 25 |
| *Experiment 2* | | |
| Col-0 | 7.9 ± 0.3 | 22 |
| *log3,4,7* | 6.3 ± 0.2* | 24 |
| *ipt3,5,7* | 9.0 ± 0.2* | 24 |
| *abcg14* | 9.2 ± 0.3* | 21 |
| *cypDM* | 11.8 ± 0.5*** | 21 |
| *CKX1ox* | 21.4 ± 0.5*** | 21 |
| *Experiment 3* | | |
| Col-0 | 8.0 ± 0.3 | 23 |
| *ckx3 ckx5* | 7.8 ± 0.3 | 23 |
| *ckx3,4,5,6* | 7.3 ± 0.2 | 23 |
| *Experiment 4* | | |
| Col-0 | 10.9 ± 0.4 | 25 |
| *arr3,4,5,6,8,9* | 10.5 ± 0.2 | 25 |
| *ahp2,3,5* | 9.7 ± 0.3** | 26 |
| *Experiment 5* | | |
| Col-0 | 8.6 ± 0.2 | 28 |
| *log3,4,7* | 7.7 ± 0.3* | 28 |
| *ahp2,3,5* | 7.9 ± 0.1* | 32 |
| *Experiment 6* | | |
| Col-0 | 8.3 ± 0.3 | 24 |
| *arr3,4,5,6,8,9* | 7.8 ± 0.3 | 25 |

Values indicate the mean number of leaves without abaxial trichomes ± SEM. Asterisks indicate significant differences compared to the wild type of the respective experiment, as calculated by Kruskal–Wallis test (experiments 1–5) or Mann–Whitney test (experiment 6) (*$q$ < 0.05; **$q$ < 0.01; ***$q$ < 0.001). Exact $p$-values and $q$-values are included in the Source Data files.

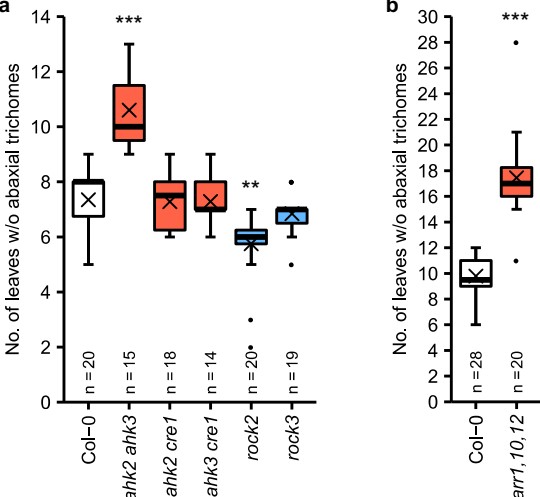

**Fig. 2 The cytokinin-dependent regulation of the juvenile-to-adult phase transition is mediated by AHK2, AHK3, ARR1, ARR10, and ARR12. a, b** Number of leaves without abaxial trichomes in cytokinin receptor mutants (**a**) and the type-B *ARR* mutant *arr1,10,12* (**b**) grown in SD. In box plots, the center line represents the median value and the boundaries indicate the 25th percentile (upper) and the 75th percentile (lower). The X marks the mean value. Whiskers extend to the largest and smallest value, excluding outliers which are shown as dots. Asterisks indicate significant differences compared to the wild type of the respective experiment, as calculated by Kruskal–Wallis test (**a**) or Mann–Whitney test (**b**) (*$p$ < 0.05; **$p$ < 0.01; ***$p$ < 0.001). Exact $p$-values are included in the Source Data files.

their involvement, we analyzed mutants of the *ARR1*, *ARR2*, *ARR10*, *ARR11*, and *ARR12* genes, as well as all their double and triple mutant combinations, since these type-B *ARR* genes have the broadest expression profiles in the shoot[52,53]. Neither the single (Supplementary Fig. 1a) nor the double type-B *ARR* mutants (Supplementary Fig. 1b) displayed any significant change in the number of juvenile leaves compared to wild-type plants. Among the triple mutants, only *arr1,10,12* produced a significantly larger number of juvenile leaves (17.5 ± 0.7 compared to 9.8 ± 0.2 juvenile leaves in wild type) (Fig. 2b), while all other combinations showed a similar number of juvenile leaves as wild type (Supplementary Fig. 1d–f). This fits to the predominant function of *ARR1*, *ARR10*, and *ARR12* in regulating vegetative shoot growth[54,55].

Further CK signaling mutants were tested: The histidine phosphotransfer protein mutant *ahp2,3,5* and the type-A *ARR* mutant *arr3,4,5,6,8,9* did not show strong differences in juvenile leaf number (Table 1).

All in all, the CK receptors AHK2 and AHK3 as well as the type-B ARRs ARR1, ARR10, and ARR12 were identified as mediators of the CK activity in the juvenile-to-adult phase transition, whereas no specificity was found for other components of the CK signaling pathway, suggesting a high functional redundancy. Modifications of the CK status in the SAM (*log4,7*; *ckx3,5*) did not cause any significant alterations of juvenile leaf number.

**CK acts downstream of miR156.** Vegetative phase change is regulated by a decrease in the abundance of the highly conserved master regulator miR156[11,12]. In order to explore whether CK acts on vegetative phase change through miR156, we measured miR156 abundance 7, 14, and 21 days after germination in shoots of SD-grown plants with a higher or lower CK status. In agreement with previously published results, we observed an overall reduction of miR156 abundance with increasing age in the wild type (Fig. 3a and Supplementary Fig. 2a)[11–13]. The same pattern and expression level were also found in genotypes with a higher or lower CK status 7 and 14 days after germination. After 21 days however, CKX1ox showed a significantly elevated miR156 level compared to the wild type (Fig. 3a). Similar results were obtained for the primary transcripts of *MIR156A* and *MIR156C* (Supplementary Fig. 2b–e), which are the most important *MIR156* genes for suppressing SPL activity during juvenile growth[13], as well as for the partly redundant miRNA miR157 (Supplementary Fig. 2f, g).

To test a possible direct short-term influence of CK on miR156 abundance, we treated 10-day-old wild-type seedlings with 1 µM 6-benzyladenine (BA) and measured the expression level at different time points after the treatment. The CK treatment caused a strong induction of several type-A *ARR* genes (Supplementary Fig. 3a), which are known CK response genes. In contrast, no significant changes in response to CK were noted neither for the abundance of miR156 (Fig. 3b) and miR157 (Supplementary Fig. 3b) nor for the transcript levels of *MIR156A* and *MIR156C* (Supplementary Fig. 3c).

Next, we crossed *ahk2 ahk3* with the miR156 target mimicry line p35S:MIM156 (MIM156). The strong expression of an artificial miR156 target in this line reduces the inhibitory effect of miR156 on *SPL* expression resulting in the absence of a juvenile phase[56,57]. Intriguingly, the *MIM156* transgene could not counterbalance the late appearance of abaxial trichomes in *ahk2 ahk3* (Fig. 3c). To exclude that this is due to transgene silencing caused by the presence of several T-DNAs in the hybrid line we

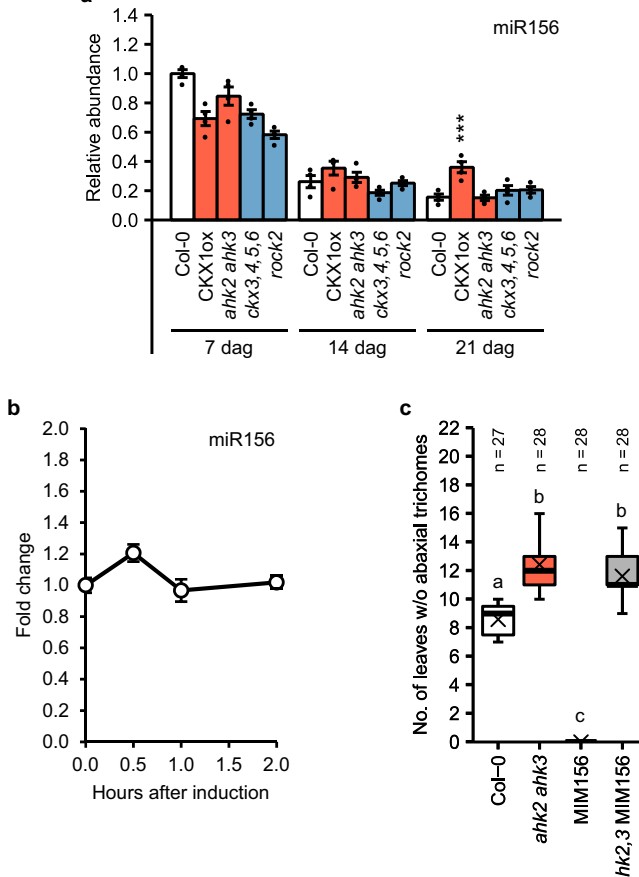

**Fig. 3 The effect of cytokinin on the juvenile-to-adult phase transition is not dependent on miR156. a** miR156 abundance in whole shoots of genotypes with an altered CK status compared to the wild type (*n* = 4 biological replicates). Dots indicate each single biological replicate. Asterisks indicate significant differences compared to the wild type of the respective time point, as calculated by one-way ANOVA, post-hoc Dunnett's test (*$p < 0.05$; **$p < 0.01$; ***$p < 0.001$). See also Supplementary Fig. 2a. **b** miR156 expression kinetics in 10-days-old SD-grown wild-type seedlings after treatment with 1 µM BA (*n* = 6 biological replicates). Statistical analysis was performed using one-way ANOVA, post-hoc Dunnett's test. No statistically significant differences were observed for miR156, comparing each time point with time point 0 ($p < 0.05$). Transcript levels (**a**, **b**) were determined by qRT-PCR. Data were normalized to *TAFII15*. Data displayed are expressed as mean ± SEM. **c** Number of leaves without abaxial trichomes in *ahk2 ahk3* MIM156 hybrid plants grown in SD. In the box plot, the center line represents the median value and the bounderies indicate the 25th percentile (upper) and the 75th percentile (lower). The X marks the mean value. Whiskers extend to the largest and smallest value, excluding outliers which are shown as dots. Letters indicate significant differences between the genotypes, as calculated by Kruskal–Wallis test ($p < 0.05$). Exact *p*-values calculated for **a**–**c** are included in the Source Data files.

determined and confirmed *MIM156* expression in *ahk2,3* MIM156 (Supplementary Fig. 4). Consistently, the transcript levels of *SPL* genes were also increased in *ahk2,3* MIM156 (Supplementary Fig. 5).

These results show that CK does not mediate its effect on vegetative phase change through miR156. In addition, the genetic analysis suggests that CK signaling is required to realize the action of miR156 on vegetative phase change, and that CK acts downstream of miR156.

**SPLs are involved in the CK-dependent regulation of the juvenile-to-adult phase transition.** The steady state mRNA levels of *SPL* genes that have been shown to play a major role in vegetative phase change[14] (*SPL2, SPL9, SPL10, SPL11, SPL13,* and *SPL15*) were similar in *ahk2 ahk3* and wild type (Supplementary Fig. 5). In order to analyze a possible immediate influence of CK on *SPL* gene expression, we measured their transcript levels in response to CK. Because a potential CK effect on *SPL* gene expression could be masked eventually in wild type by the counteraction of miR156, we treated the MIM156 line with BA. Type-A *ARR* expression in MIM156 in response to CK was determined as an induction control, which showed the expected increase (Supplementary Fig. 6). In contrast, no significant changes of *SPL* transcript levels were detected after CK treatment (Fig. 4a).

To investigate further the potential role of SPLs in CK-regulated phase transition, we carried out a genetic analysis and introgressed the early-transitioning *rock2* mutation into the well-characterized *spl9 spl15* mutant[58]. The *rock2* mutation did not rescue the delayed vegetative phase change of *spl9 spl15* (Fig. 4b), suggesting that *SPL9* and *SPL15* are not or not the only *SPL* genes mediating the CK effect. Supporting this conclusion, the *rock2* mutation reduced the percentage change of juvenile leaf number by a similar portion in both backgrounds (about 18% in wild type and 16% in *spl9 spl15*). To extend the analysis on all miR156-targeted *SPL* genes, we introgressed the *rock2* mutation into a transgenic line expressing *p35S:MIR156B* (MIR156ox), which allows no accumulation of *SPL* transcripts resulting in a substantially prolonged juvenile phase[11,17,23,59]. *Rock2* was not able to accelerate the transition to the adult phase caused by *p35S:MIR156B* expression (Fig. 4c), indicating that regulation of vegetative phase change by CK relies on SPL function.

**The influence of CK on vegetative phase change depends on miR172 as well as TOE1 and TOE2.** SPLs are positive regulators of miR172 expression and consequently, the age-dependent increase of SPL levels causes a progressive elevation of miR172 abundance in wild-type plants[11,12,18,21] (Supplementary Fig. 7a). An increase of miR172 with advancing age was also visible in plants with an altered CK status (Fig. 5a). However, it progressed slower in CK-deficient plants and faster in plants with a higher CK status. Small differences in miR172 abundance were already detectable 7 days after germination and became more prominent at later time points. Twenty-one days after germination only about half of the miR172 level of wild type was present in plants with a lower CK status. In contrast, plants with a higher CK status showed a higher miR172 level (Fig. 5a). Among the tested miR172 precursors, *MIR172A* and *MIR172B*, which were shown to play dominant roles in the timing of trichome initiation[24], showed a similar expression pattern (Supplementary Fig. 7b–e).

Next, we tested the response of *MIR172* gene expression and mature miR172 levels to CK. BA treatment resulted in an up to two-fold increase of *MIR172A* and *MIR172B* transcript levels within 1–2 h (Fig. 5b). The response of *MIR172* gene expression to CK was also reflected by transiently increased mature miR172 levels upon CK treatment (Fig. 5c). In order to test the dependence of the CK-mediated regulation of vegetative phase change on miR172, we crossed *ahk2 ahk3* with a *MIR172B* overexpression line (MIR172ox). For unknown reasons, we were unable to generate a homozygous hybrid line and also direct transformation of the transgene into the *ahk2 ahk3* receptor mutant did not yield any primary transformants. However, transforming *p35S:MIR172B* into the less severe *ahk2-5 ahk3-7* mutant[60] yielded transformants that were directly used for phenotyping under SD conditions. Since control lines could not

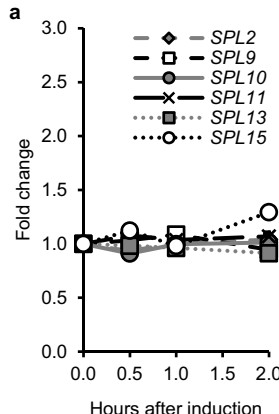

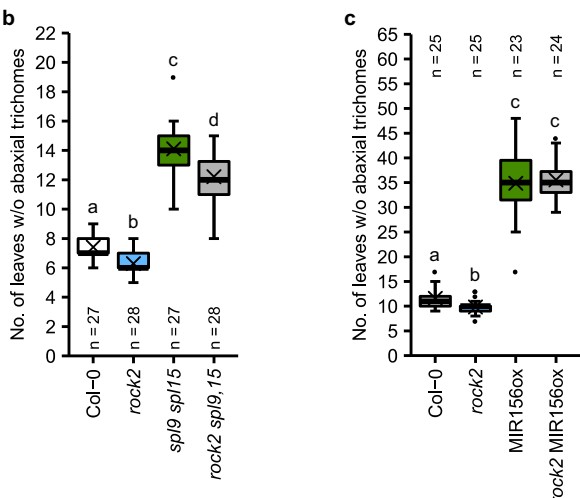

**Fig. 4 SPLs are involved in the cytokinin-dependent regulation of vegetative phase change. a** *SPL* expression kinetics in 10-day-old SD-grown MIM156 seedlings after treatment with 1 μM BA. Transcript levels were determined by qRT-PCR. Data were normalized to *TAFII15* and *PP2AA2*. Data displayed are expressed as mean ± SEM ($n = 6$ biological replicates). Statistical analyses were performed using one-way ANOVA, post-hoc Dunnett's test. No statistically significant differences were observed for any *SPL* gene, comparing each time point with time point 0 ($p < 0.05$). **b**, **c** Number of leaves without abaxial trichomes in *rock2 spl9 spl15* plants (**b**) and *rock2* MIR156ox plants (**c**) grown in SD. In box plots, the center line represents the median value and the boundaries indicate the 25th percentile (upper) and the 75th percentile (lower). The X marks the mean value. Whiskers extend to the largest and smallest value, excluding outliers which are shown as dots. Letters indicate significant differences between the genotypes, as calculated by Kruskal–Wallis test ($q < 0.05$). Exact $p$-values and $q$-values calculated for **a**–**c** are included in the Source Data files.

be grown on kanamycin-containing media in this experiment, a control experiment was conducted separately, comparing Col-0 and *ahk2-5 ahk3-7* with respective T1 plants harboring the empty vector, showing no impact of selective media on juvenile leaf number (Supplementary Fig. 8). Transgene expression and presence of mature miR172 was confirmed by qRT-PCR (Supplementary Fig. 9). Similar to *ahk2-2tk ahk3-3* used for the other experiments, *ahk2-5 ahk3-7* showed a reduction of *MIR172B* expression and miR172 abundance (Fig. 5a and Supplementary Figs. 7e and 9). *Ahk2-5 ahk3-7* also displayed a late-transitioning phenotype which was completely suppressed by the *p35S:MIR172B* transgene (Fig. 5d), supporting the hypothesis

of CK influencing vegetative phase change by regulating miR172 expression.

The regulation of *MIR172* expression by CK should have an impact on miR172 targets as well. Since the miR172-mediated silencing of *AP2*-like genes occurs mainly on the translational level[18,19,61], we refrained from measuring transcript levels and conducted a genetic analysis instead. The mutation of *SMZ* did not alter juvenile leaf number, but reduced it in the *ahk2 ahk3* background (Supplementary Fig. 10a), suggesting a role for *SMZ* in the CK-dependent regulation of vegetative phase change. Simultaneous loss of *SMZ* and its close homolog *SNZ* caused a slightly earlier transition to the adult phase in the wild type. Introgression of both mutant alleles into the *ahk2 ahk3* background resulted also in an earlier juvenile-to-adult phase transition with the effect being larger than in wild type (Fig. 6a). This indicates that SMZ and SNZ are required for regulation of the juvenile-to-adult phase transition in plants with a lower CK status but that additional factors play a role.

Loss of single *TOE1* or *TOE2* gene functions in the background of *ahk2 ahk3* also showed their participation in mediating the CK activity (Supplementary Fig. 10b). The juvenile leaf number of *ahk2,3 toe1,2* plants was indistinguishable from *toe1 toe2* control plants (Fig. 6b). The additional disruption of the *TOE3* gene in the *ahk2 ahk3* background did not result in a further reduction of the number of juvenile leaves (Supplementary Fig. 10c). Furthermore, introgression of *rock2* into the *toe1 toe2* mutant did not result in a further decrease in juvenile leaf number (Fig. 6c) showing a strong epistatic relationship. These findings strongly suggest that these two miR172 target genes take part in the CK-dependent regulation of vegetative phase change.

## Discussion

This work has revealed a pivotal role of CK in the control of vegetative phase change in *Arabidopsis*. Several key factors mediating the influence of CK on the juvenile-to-adult phase transition were identified and a link of the hormone to components of the age pathway is described (Fig. 7).

IP and *t*Z are generally considered to be the most active natural CKs, whereas *c*Z has mostly a lower activity[62]. Analysis of CK biosynthesis and transport mutants showed the importance of *t*Z-type CKs in the regulation of vegetative phase change. The *abcg14* mutant, which is impaired in the long-distance allocation of *t*Z-type CKs from the root to the shoot[63,64], as well as the *t*Z biosynthesis mutant *cypDM*[65] produced ~15% and ~30% more juvenile leaves than the wild type, respectively (Table 1). Both mutants compensate the lack of *t*Z at least in part by increased production of iP-type CKs[63–65], but this was not sufficient to allow proper timing of vegetative phase change, suggesting a less important role for iP in this process. Moreover, the vegetative phase change in the *ipt2,9* mutant was similar to wild type indicating that *c*Z-type CKs have no role in this process. Notably, the significant retardation of the transition to the adult phase in the *abcg14* mutant established CK as a root-borne factor promoting vegetative phase change, which was shown previously to be initiated mainly in the SAM as well as in the leaves themselves[66,67]. However, compared to its role in promoting the transition to flowering under SD[68,69] the root-derived portion of *t*Z appears to have a more limited role in regulating vegetative phase change.

The more important function of *t*Z-type CKs in the juvenile-to-adult phase transition compared to iP-type CKs is in agreement with the CK affinities of the involved receptors AHK2 and AHK3. AHK3 has been shown to display a higher affinity to *t*Z than to iP[70] and the *in planta* CK response mediated by AHK2 and AHK3 is triggered more strongly by *t*Z than by iP[71].

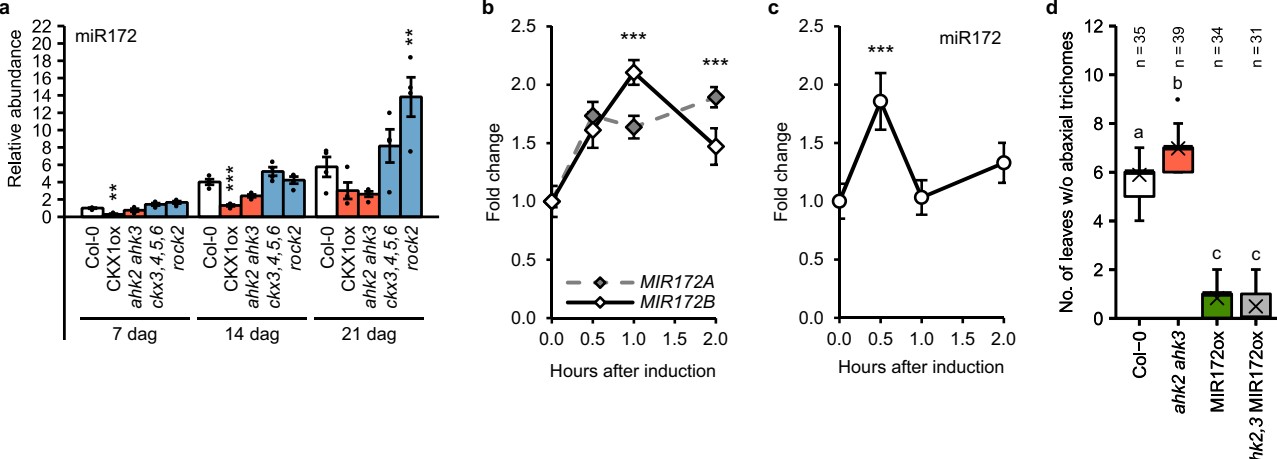

**Fig. 5 Cytokinin increases miR172 abundance. a** miR172 abundance in whole shoots of genotypes with an altered CK status compared to the wild type (n = 4 biological replicates). See also Supplementary Fig. 7a. Dots indicate each single biological replicate. **b, c** miR172 and *MIR172* gene expression kinetics in 10-day-old SD-grown wild-type seedlings after treatment with 1 μM BA (n = 6 biological replicates). Transcript levels were determined by qRT-PCR. Data were normalized to *TAFII15* (**a, c**) or *TAFII15* and *PP2AA2* (**b**). Data displayed are expressed as mean ± SEM. Asterisks indicate significant differences compared to the wild type of the respective time point (**a**) or compared to time point 0 (**b, c**), as calculated by one-way ANOVA, post-hoc Dunnett's test (*$p < 0.05$; **$p < 0.01$; ***$p < 0.001$). **d** Number of leaves without abaxial trichomes in *ahk2-5 ahk3-7* MIR172ox hybrid plants grown in SD. T1 plants shown in Supplementary Fig. 9 were used for the analysis. In box plots, the center line represents the median value and the boundaries indicate the 25th percentile (upper) and the 75th percentile (lower). The X marks the mean value. Whiskers extend to the largest and smallest value, excluding outliers which are shown as dots. Letters indicate significant differences between the genotypes, as calculated by Kruskal–Wallis test (q < 0.05). Exact *p*-values and *q*-values calculated for **a**–**d** are included in the Source Data files.

Downstream of these receptors the type-B ARRs ARR1, ARR10, and ARR12 were identified to mediate the influence of CK on the juvenile-to-adult phase transition (Fig. 2b), which have been described previously as central regulators of shoot growth[54,55,72]. Notably, ARR2 has apparently no role (Supplementary Fig. 1d–f) despite its known functions in leaves[73–75] where it mediates the inhibition of senescence by CK at a later developmental stage[76]. Taken together, the genetic analysis has shown the relevance of *t*Z-type CK for the juvenile-to-adult phase transition acting through the AHK2/AHK3-ARR1/ARR10/ARR12 signaling module.

Further analysis uncovered the participation of distinct components of the age pathway in mediating the action of CK. Transcriptional and genetic analysis argues against the involvement of miR156 in that process. MiR156 and also miR157 levels were not altered in CK mutants in early developmental stages or by exogenously applied CK (Fig. 3a, b and Supplementary Figs. 2g and 3b). However, CK signaling was required to induce an early transition to the adult phase in the absence of the repressive master regulator miR156: Introgression of the late-transitioning *ahk2 ahk3* double mutant into the MIM156 line completely abolished the effect of the target mimicry on the phenotype, while not substantially impacting the expression of the *SPL* transcript abundance in the hybrid line (Fig. 3c and Supplementary Fig. 5). It is evident, that CK does not affect vegetative phase change miR156-dependently, but acts downstream of it.

On the other hand, we showed that CK regulates the vegetative phase change miR172-dependently: MiR172 abundance positively correlated with the CK status in different CK mutants and increased transiently after CK treatment (Fig. 5a, c). Additionally, the *ahk2 ahk3* mutant was unable to counteract the complete loss of the juvenile phase caused by expressing a *p35S:MIR172B* transgene (Fig. 5d).

No changes in expression of miR156/miR157-regulated *SPL* genes were observed in the CK receptor mutant *ahk2 ahk3* (Supplementary Fig. 5) or by treatment with exogenous CK (Fig. 4a). However, the inability of the *rock2* mutation to

counteract the effect of *MIR156B* overexpression (Fig. 4c) implies an involvement of SPLs in the CK-dependent regulation of vegetative phase change. *Arabidopsis* has 16 *SPL* genes, ten of which are targeted by miR156/miR157[13,58,77,78]. At least five of the SPLs induce *MIR172* gene expression: SPL2, SPL9, SPL11, SPL13A/B, and SPL15[11,14]. CK has a positive effect on miR172 abundance as well (Fig. 5a–c and Supplementary Fig. 7c, e). These activities could be independent events, but it is also possible that SPLs and type-B ARRs or other CK signaling components induce *MIR172* expression in a cooperative fashion. Both SPLs and type-B ARRs bind to *MIR172* gene loci[11,79–81]. Furthermore, Zhang et al.[82] showed physical interaction of SPL9 with several type-B ARRs, including ARR1, ARR10, and ARR12, causing a reduction of the CK-dependent shoot regeneration capacity. This repression of CK activity by SPLs contrasts with their aligned activities in promoting the juvenile-to-adult phase transition. But it could be that the consequences of the SPL-ARR interaction are context-specific. The molecular mechanism of CK action through SPLs remains to be shown.

Yant et al.[25] showed that all six miR172 target genes have to be mutated in order to mimic a miR172-overexpressing plant. Among the target genes, *TOE1* and *TOE2* have the strongest influence on the length of the juvenile phase, since mutating them reduces juvenile leaf number by more than half[11] (Fig. 6b, c). Not much is known about the degree of influence of the other four genes. Loss of *SMZ* and *SNZ* only slightly reduced the number of juvenile leaves (Fig. 6a and Supplementary Fig. 10a) and *AP2* and *TOE3* are mostly known for their role in flower development[21,83]. The knockout mutants *toe1*, *toe2*, *smz*, and *smz snz* all showed partial restoration of the juvenile phase of *ahk2 ahk3* with *toe2* having the strongest impact (Fig. 6a and Supplementary Fig. 10a, b), whereas loss of both *TOE1* and *TOE2* completely suppressed the late transition of *ahk2 ahk3* (Fig. 6b), indicating that *TOE1* and *TOE2* are both necessary and sufficient for the CK-regulation of the juvenile-to-adult phase transition.

Taken together, we added a regulatory layer to the age-dependent control of vegetative phase change, with CK

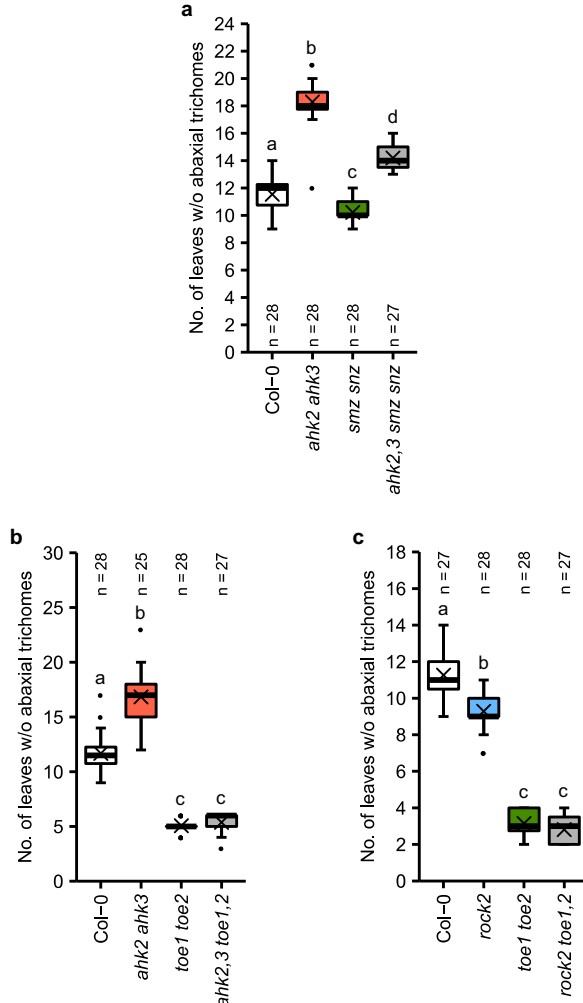

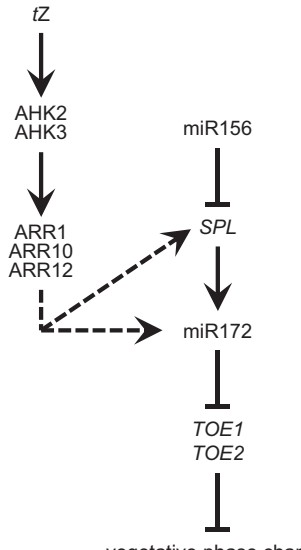

**Fig. 7 Model of the cytokinin-dependent regulation of vegetative phase change.** The CK receptors AHK2 and AHK3 as well as the type-B response regulators ARR1, ARR10, and ARR12 are activated mainly by tZ. Activation of CK signaling in the shoot results in an increase of miR172 abundance. MiRNA activation may be achieved directly by type-B ARRs or involve co-action of SPLs. TOE1 and TOE2 are repressed posttranscriptionally upon miR172 increment, enabling the transition from the juvenile to the adult vegetative phase.

**Fig. 6 Genetic interactions between *AP2-like* genes and cytokinin receptor genes regarding vegetative phase change.** Number of leaves without abaxial trichomes in *ahk2,3 smz snz* (**a**), *ahk2,3 toe1,2* (**b**) and *rock2 toe1,2* (**c**) hybrid plants grown in SD. In box plots, the center line represents the median value and the boundaries indicate the 25th percentile (upper) and the 75th percentile (lower). The *X* marks the mean value. Whiskers extend to the largest and smallest value, excluding outliers which are shown as dots. Letters indicate significant differences between the genotypes, as calculated by Kruskal–Wallis test ($q < 0.05$). Exact *q*-values are included in the Source Data files.

promoting *MIR172* gene expression (most probably in cooperation with SPLs), while not affecting the other key regulator miR156. Additionally, we identified TOE1 and TOE2 as being specifically important in the CK-dependent regulation of this process (Fig. 7).

Fouracre and Poethig[67] proposed that miR156 or a miR156-derived signal produced by the SAM ensures juvenility of the first leaves, but as the plant grows, the identity of subsequent leaves is determined by peripheral organs to an increasing degree. Since CK acts on miR172 rather than miR156, CK would only affect later formed leaves, supporting the hypothesis that CK regulates vegetative phase change together with SPLs, whose expression is increased in later formed leaves when miR156 abundance decreases. Hence, CK might not be able to influence the identity of the first leaves, which explains why none of the tested mutants with a higher CK status show a reduction of juvenile leaf number as strong as *toe1 toe2* or even p35S:MIR172 or p35S:MIM156

plants. This hypothesis is supported by the fact that all CK signaling components identified in this study to be involved in regulating vegetative phase change are highly expressed in leaves[52,53,84]. Furthermore, the importance of CK activity in the SAM for vegetative phase change is low, since loss of CK genes known to be active in the SAM (*CKX3* and *CKX5*[51]; *LOG4* and *LOG7*[50]) did not cause any alterations in juvenile leaf number (Table 1).

Future work should explore the functional relevance of CK under different environmental conditions. A predominant activity of CK in regulating the juvenile-to-adult transition in later formed leaves would be consistent with the idea that the hormone mediates responses to environmental cues[27], which are partly known to regulate the juvenile-to-adult phase transition[85–87] to adapt to environmental changes[88]. Additionally, the question should be addressed whether other leaf traits characterizing vegetative phase change[6] respond with similar sensitivity to an altered CK status as does epidermal identity analyzed here. MiR156 inhibits the development of all adult traits, with different SPLs promoting different subsets of these traits[11], while TOE1 and TOE2 are only involved in the formation of abaxial trichomes[11,25,89]. Furthermore, the nature of the ARR-SPL interaction remains to be clarified, which might uncover other age-related processes regulated by the interplay of the two pathways.

### Methods
**Plant material and growth conditions**. The Columbia-0 (Col-0) ecotype of *Arabidopsis thaliana* was used as the wild type. All mutants and transgenic lines that were used in this study and generated by crossings are listed in Supplementary Table 1. As a *smz* mutant, we used a previously uncharacterized knock-out allele (SALK_135824C) which was named *smz-4*. No full-length *SMZ* transcript was detected in this mutant by semi-quantitative RT-PCR, suggesting that this is a null allele (Supplementary Fig. 11). Primers used for the analysis of *smz-4* are listed in Supplementary Table 2. The mutant alleles used for the generation of the *ckx3,4,5,6* quadruple mutant were described in Bartrina et al.[51]. All genotypes were

propagated under LD conditions (16 h dark/8 h light cycle), 22 °C and 30–65% humidity, and confirmed by PCR analysis. Primers used for genotyping are listed in Supplementary Table 3. For the analyses of juvenile leaf number and the quantification of miRNA abundance and gene expression in CK mutants, *Arabidopsis* plants were grown on soil with a 8 h light/16 h dark cycle, at 22 °C and 60% humidity and light intensities of 100–150 μmol m$^{-2}$ s$^{-1}$. *Ahk2,3* MIR172ox T1 plants were selected on ½ MS agar plates (0.22% (w/v) MS basal salt, 0.05% (w/v) MES, 0.5% (w/v) sucrose, 0.8% (w/v) agar, pH 5.7) containing 30 μg/ml kanamycin and transferred to soil after 14 days. Pots of different lines were randomized by default to minimize positional effects.

**Plasmid constructions for transgenic plants**. For the generation of the *p35S:MIR172B* construct, the Gateway® system (ThermoFisher, Waltham, MA) was used. All primers used for cloning purposes are listed in Supplementary Table 4. A *35S* promoter fragment of the Cauliflower Mosaic Virus was PCR amplified from the vector pGWB15[90] using the primers p35S-F-attB4 and p35S-R-attB1r, and recombined into the donor vector pDONR$^{TM}$P4-P1R using Gateway® BP Clonase® Enzyme Mix (ThermoFisher, Cat. No. 11789013). *MIR172B* was amplified from Col-0 genomic DNA using the primers MIR172B-attB1-2_fw and MIR172B-attB2-2_rv and inserted into pDONR$^{TM}$221 via BP reaction. The *MIR172B* sequence was combined with the *35S* promoter in pK7m24GW[91] to create the *p35S:MIR172B* construct. The multisite LR reaction was performed with the Gateway® LR Clonase® II Enzyme Mix (ThermoFisher, Cat. No. 11791020). The cloned gene construct was fully sequenced to ensure that no mutation was introduced. The plasmid was transformed into the *Agrobacterium tumefaciens* strain GV3101 by electroporation and the resulting bacterial strain was used to transform *Arabidopsis* plants using the floral-dip method[92].

**CK induction assays**. For the determination of the influence of CK on gene expression, seeds were surface-sterilized using a 1.2% (v/v) sodium hypochlorite/0.01% (v/v) Triton X-100 solution. Seedlings were grown under SD conditions for 10 days in liquid ½ MS medium (0.22% (w/v) MS basal salt, 0.05% (w/v) MES, 0.1% (w/v) sucrose, pH adjusted to 5.7). 6-Benzylaminopurine (BA) was dissolved in 1 M KOH. As a control, 1 M KOH was used. Both solutions were diluted in 0.05 (w/v) MES and the pH was adjusted before adding them to the medium. CK application and harvesting of plant material at the different time points was conducted during the night, starting 1.5 h after its beginning.

**RNA preparation and quantitative RT-PCR**. Approximately 100 mg of plant material was harvested and frozen in liquid nitrogen at the indicated time points. The frozen samples were ground using a Retsch mill in precooled adapters. Total RNA was extracted using TRIsure$^{TM}$ (Bioline) following the manufacturer's instructions. Eighty percent (v/v) ethanol was used to wash the RNA pellet, which was resuspended in 40–50 μl of nuclease-free water and treated with DNase I (ThermoFisher) following the manufacturer's instructions. For normal cDNA synthesis, 1–1.5 μg of total RNA was reversely transcribed using SuperScript$^{TM}$ III (ThermoFisher), 4.5 μM of N9 random oligos and 2.5 μM of oligo-dT$_{25}$ in a 20 μl reaction. Mix 1 containing RNA, 2 mM of dNTP mix and oligos was incubated for 5 min at 65 °C and placed on ice afterwards. Mix 2 (first strand buffer, 5 mM DTT, SuperScript$^{TM}$ III) was added and samples were incubated for 30 min at 25 °C, 60 min at 50 °C, and 15 min at 70 °C. The resulting cDNA was diluted 1:5. For detection of mature miRNAs, mix 1 contained 500 ng of total RNA, 1 mM of dNTP mix, 25 nM TAFII15-StLp-cDNA_rv as an internal control for qRT analysis, and 25 nM of the respective miRNA-specific stem-loop primer (Supplementary Table 5). Stem-loop RT primers were designed according to Chen et al.[93]. Following the addition of mix 2 containing first strand buffer, 4 mM DTT, 0.6 U/μl RNaseOUT (ThermoFisher), SuperScript$^{TM}$ III and resulting in a total volume of 12.5 μl, samples were incubated for 30 min at 16 °C, 30 min at 50 °C, and 15 min at 70 °C. Undiluted cDNA was used in the qRT-PCR reactions. For qRT-PCR analyses, *PROTEIN PHOSPHATASE 2A SUBUNIT A2 (PP2AA2)* and *TBP-ASSOCIATED FACTOR II 15 (TAFII15)* served as reference genes. All qRT-PCR primers used in this study are listed in Supplementary Table 5. qRT-PCR was performed with the CFX96$^{TM}$ Real-Time Touch System (Bio-Rad®) using SYBR Green I as DNA-binding dye. Gene expression data analysis was carried out according to Vandesompele et al.[94]. For analysis of *MIM156* transgene expression, the 40-ΔCt method was used, as described by Morcuende et al.[95].

**Statistical analysis**. Statistical analyses were performed using GraphPad Prism, version 8 (GraphPad Software, La Jolla, CA). Statistical tests used were all two-sided and are indicated in the figure and table legends. The data was analyzed by one-way analysis of variance (ANOVA) followed by Dunnett's or Tukey's post hoc test, Kruskal–Wallis test followed by either Dunn's test or two-stage step-up procedure of Benjamini, Krieger, and Yukitieli, or two-tailed Mann-Whitney test. A *p*-value or *q*-value < 0.05 was considered to indicate a statistically significant difference. In case of transcript analyses, a 1.75-fold upregulation or downregulation compared to the respective control was chosen as threshold.

**Reporting summary**. Further information on research design is available in the Nature Research Reporting Summary linked to this article.

## Data availability
Source data and statistical information are provided with this paper.

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

## Acknowledgements
We acknowledge excellent technical assistance by Gabriele Grüschow and thank Tim Krahn and Jutta Hoffmann for their contributions made during their bachelor thesis. We thank Youngsook Lee and the Nottingham Arabidopsis Stock Centre for providing seeds. The vector pDONR™P4-P1R/*p35S* was kindly provided by Elisabeth Otto.

## Author contributions
S.W. discovered the influence of cytokinin on vegetative phase change; S.W., I.B., and T.S. developed the project; S.W. performed experiments; S.W., I.B., and T.S. analyzed data; S.W. and T.S. wrote the article, I.B. read and contributed to previous versions and approved the final version.

## Funding

## Competing interests
The authors declare no competing interests.
