## [Peer Review File · Nature Communications]

Cytokinin regulates vegetative phase change in Arabidopsis through the miR172/TOE1-TOE2 moduleREVIEWER COMMENTS

Reviewer #1 (Remarks to the Author):

The hormone regulation of vegetative phase transition is poorly understood. Previous studies have revealed that gibberellin regulates the juvenile-to-adult transition and flowering time in many plant species. The manuscript presented here demonstrate that cytokinin (CK), another important phytohormone, also influences the length of juvenile phase in Arabidopsis. Therefore, the topic of this paper is interesting and contributes to our understanding of vegetative phase transition in plants.

The conclusion of this paper is mainly drawn by extensive gene expression and genetic analyses. Based on the phenotypic analyses of diverse CK-related mutants, the most interesting result of this manuscript is that the root-derived tZ significantly influenced the phase transition. Overall, the paper is well written and the results are clearly presented. A few concerns should be solved before I can recommend it for publication in Nat Commun.

1. My major concern is how to precisely define the length of juvenile phase in the CK-related mutants. As authors stated at the end of their manuscript, whether other leaf traits characterizing vegetative phase change respond with similar sensitivity to an altered CK status is currently unknown. Therefore, I am not completely convinced that the timing of abaxial trichome formation is sufficient to define the transition point when the plants transit from juvenile to adult phase. On the basis of the definition, the adult phase differs from the juvenile phase in terms of reproductive competence. Is it possible to measure the flowering time of these mutants? Will it give rise to consistent results? This point should be clarified.

2. For qRT-PCR analysis: How did authors measure the levels of mature miR156 and miR172 by qRT-PCR? I could not find any information in the Method. In addition, Why PROTEIN PHOSPHATASE 2A SUBUNIT A2 (PP2AA2) and TBP-ASSOCIATED FACTOR II 15 (TAFII15) were used as reference genes? Any particular reason for this? How did authors harvest samples for qRT-PCR at day 7, 14 and 21 (e.g. Fig. 3a and Fig. 5a)? Did they use all the aerial parts? Or they just harvested some rosette leaves? Is there any difference among different leaves? These points should be also clarified.

Reviewer #2 (Remarks to the Author):

The authors here present a strong line of genetic evidence that demonstrates that Cytokinin signalling mediates the vegetative phase change by interacting with the mi172 pathway. This provides us with new insights on the juvenile to adult phase transition. This work capitalises on many mutants and lines to mechanistically interfere, and the methodology is sound and thorough.

concerns:

line 192 to 200: the logic seems to me a bit lost. What about inverting the arguments and presenting first that rock 2 has an effect in the double spl9 spa 115, indicating that other SPL genes might be involved, and then presenting the experiment 35S:MIR156B line.

Line 287 I would argue that this is a positive correlation, in the mutants with low signalling it is low and it is increased upon CK stimulation

REVIEWER COMMENTS

Reviewer #1 (Remarks to the Author):

The hormone regulation of vegetative phase transition is poorly understood. Previous studies have revealed that gibberellin regulates the juvenile-to-adult transition and flowering time in many plant species. The manuscript presented here demonstrate that cytokinin (CK), another important phytohormone, also influences the length of juvenile phase in Arabidopsis. Therefore, the topic of this paper is interesting and contributes to our understanding of vegetative phase transition in plants.

The conclusion of this paper is mainly drawn by extensive gene expression and genetic analyses. Based on the phenotypic analyses of diverse CK-related mutants, the most interesting result of this manuscript is that the root-derived tZ significantly influenced the phase transition. Overall, the paper is well written and the results are clearly presented. A few concerns should be solved before I can recommend it for publication in Nat Commun.

1. My major concern is how to precisely define the length of juvenile phase in the CK-related mutants. As authors stated at the end of their manuscript, whether other leaf traits characterizing vegetative phase change respond with similar sensitivity to an altered CK status is currently unknown. Therefore, I am not completely convinced that the timing of abaxial trichome formation is sufficient to define the transition point when the plants transit from juvenile to adult phase. On the basis of the definition, the adult phase differs from the juvenile phase in terms of reproductive competence. Is it possible to measure the flowering time of these mutants? Will it give rise to consistent results? This point should be clarified.

Answer

Flowering time has been analyzed for all mutants shown in this manuscript and the results will be published soon. It is shown that cytokinin has also a positive influence on the transition to the reproductive phase. Mostly the same cytokinin genes are involved in regulating vegetative phase change and the transition to flowering and at least partly different components of the age pathway play a role in realizing the cytokinin output in both developmental transitions. However, flowering time might not be the best feature to support our data since it is not only influenced by age, and the age pathway might not even be the most important one to regulate the transition to reproductive growth. In that respect we refer to Poethig, 2013 (doi:10.1016/B978-0-12-396968-2.00005-1), who wrote: „[...] genetic analyses of vegetative phase change and floral induction indicate that these developmental transitions are inherited independently in Eucalyptus (Jordan et al., 1999; Wiltshire et al., 1998), Pisum (Wiltshire et al., 1994), maize (Abedon et al., 1996), and Arabidopsis (Telfer et al., 1997).”

It is indeed the question whether there might be a differential effect of cytokinin of different traits of the juvenile-to-adult transition as is indicated in the discussion. However, analyzing the appearance of abaxial trichomes is used by most if not all authors to determine the end of the juvenile phase, so we adapted this community standard for our analyses. In some articles, the length:width ratio of the leaf blade or the angle of the leaf base is shown in addition. Figure 1 shows that CK-deficient plants have more leaves that are round-shaped and unserrated indicating that other traits than appearance of abaxial trichomes might be altered too. Unfortunately, no unequivocal molecular markers are known to study the juvenile-to-adult transition.

2. For qRT-PCR analysis: How did authors measure the levels of mature miR156 and miR172 by qRT-PCR? I could not find any information in the Method. In addition, Why PROTEIN PHOSPHATASE 2A

SUBUNIT A2 (PP2AA2) and TBP-ASSOCIATED FACTOR II 15 (TAFII15) were used as reference genes? Any particular reason for this? How did authors harvest samples for qRT-PCR at day 7, 14 and 21 (e.g. Fig. 3a and Fig. 5a)? Did they use all the aerial parts? Or they just harvested some rosette leaves? Is there any difference among different leaves? These points should be also clarified.

Answer

The method to measure levels of mature miR156 and miR172 is described in the Methods section, lines 391-397. cDNA syntheses were performed using miRNA-specific stem-loop primers (designed according to Chen et al., 2005) and a primer for reversely transcribing the reference gene. The stem-loop qRT-PCR protocol was modified from Chen et al. (2005) and SYBR Green was used as the fluorescence dye rather than TaqMan probes. All primers are listed in Supplementary Tab. 5.

Several potential reference genes were tested and these two turned out to be the most stable ones in our experiments. These two reference genes have been used in several published articles, e. g. Cortleven et al., 2019 (doi:10.1093/jxb/ery344), Frank et al., 2020 (doi:10.1111/pce.13860), Bursch et al., 2020 (doi:10.1038/s41477-020-0725-0), Werner et al., 2021 (doi:10.3389/fpls.2021.613488).

Whole shoots were harvested for qRT-PCR analyses, meaning all aerial parts of the plants including the meristem and the leaves. This information is now included in the respective figure legends (3a, 5a, S2, S7).

Age pathway components are not evenly expressed in the whole shoot, e. g. miR156 and miR157 abundances are highest in the first two leaves and decrease significantly from leaves 1&2 to leaves 3&4. In successive leaves the decline is more gradual, reaching a relatively constant level (He et al., 2018, doi:10.1371/journal.pgen.1007337). In addition, there is an age-dependent decrease of miR156, leading to an increase in miR172 abundance. By analyzing the whole shoot we aimed to obtain a general picture of the impact of the cytokinin status on the abundance of these two miRNAs. This analysis was complemented by the induction assay .

Reviewer #2 (Remarks to the Author):

The authors here present a strong line of genetic evidence that demonstrates that Cytokinin signalling mediates the vegetative phase change by interacting with the mi172 pathway. This provides us with new insights on the juvenile to adult phase transition. This work capitalises on many mutants and lines to mechanistically interfere, and the methodology is sound and thorough.

concerns:

line 192 to 200: the logic seems to me a bit lost. What about inverting the arguments and presenting first that rock 2 has an effect in the double spl9 spa 115, indicating that other SPL genes might be involved, and then presenting the experiment 35S:MIR156B line.

Answer

We have changed the order of arguments as suggested and therefore exchanged Fig. 4b and 4c.

Line 287 I would argue that this is a positive correlation, in the mutants with low signalling it is low and it is increased upon CK stimulation

Answer

This is correct. „negatively“ was changed to „positively“

REVIEWERS' COMMENTS

Reviewer #1 (Remarks to the Author):

The response to my concerns sound reasonable. Therefore, I am happy to recommend it for publication in Nat Commun.